# An Instrumented Mouthguard for Real-Time Measurement of Head Kinematics under a Large Range of Sport Specific Accelerations

**DOI:** 10.3390/s23167068

**Published:** 2023-08-10

**Authors:** Chris M. Jones, Kieran Austin, Simon N. Augustus, Kieran Jai Nicholas, Xiancheng Yu, Claire Baker, Emily Yik Kwan Chan, Mike Loosemore, Mazdak Ghajari

**Affiliations:** 1Sports and Wellbeing Analytics, Swansea SA7 0AJ, UK; kieran.austin@swa.one (K.A.);; 2Institute of Sport and Exercise Health (ISEH), Division Surgery Interventional Science, University College London, London W1T 7HA, UK; 3Institute of Sport, Nursing and Allied Health, University of Chichester, Chichester PO19 6PE, UK; 4Department of Applied and Human Sciences, Kingston University London, London KT1 2EE, UK; 5HEAD Lab, Dyson School of Design Engineering, Imperial College London, South Kensington Campus, London SW7 2AZ, UK; x.yu16@imperial.ac.uk (X.Y.);; 6English Institute of Sport, Manchester M11 3BS, UK

**Keywords:** piezoelectric sensors, gyroscopes, mouthguard, head kinematics, traumatic brain injury, sporting impacts

## Abstract

Background: Head impacts in sports can produce brain injuries. The accurate quantification of head kinematics through instrumented mouthguards (iMG) can help identify underlying brain motion during injurious impacts. The aim of the current study is to assess the validity of an iMG across a large range of linear and rotational accelerations to allow for on-field head impact monitoring. Methods: Drop tests of an instrumented helmeted anthropometric testing device (ATD) were performed across a range of impact magnitudes and locations, with iMG measures collected concurrently. ATD and iMG kinematics were also fed forward to high-fidelity brain models to predict maximal principal strain. Results: The impacts produced a wide range of head kinematics (16–171 g, 1330–10,164 rad/s^2^ and 11.3–41.5 rad/s) and durations (6–18 ms), representing impacts in rugby and boxing. Comparison of the peak values across ATD and iMG indicated high levels of agreement, with a total concordance correlation coefficient of 0.97 for peak impact kinematics and 0.97 for predicted brain strain. We also found good agreement between iMG and ATD measured time-series kinematic data, with the highest normalized root mean squared error for rotational velocity (5.47 ± 2.61%) and the lowest for rotational acceleration (1.24 ± 0.86%). Our results confirm that the iMG can reliably measure laboratory-based head kinematics under a large range of accelerations and is suitable for future on-field validity assessments.

## 1. Introduction

Contact and collision sports such as boxing and rugby expose athletes to traumatic brain injuries as a result of head impacts [1]. Recent evidence has started to relate head impact exposure to negative long-term effects such as the neurodegenerative disease chronic traumatic encephalopathy (CTE) [2,3,4,5,6]. However, there is still a lack of real-world traumatic brain injury (TBI) data paired with head impact kinematics during contact sports. Such data will allow for exploring the links between head impact exposure and TBI and improving injury surveillance systems.

New sensing technologies have led to the development of smart wearables that can measure head kinematics, such as linear acceleration, rotational acceleration, or rotational velocity. Among all wearables, instrumented mouthguards (iMG) are thought to provide the most accurate measurement of the head kinematics because they are fitted to the upper dentition, hence closely attached to the skull [7,8,9,10]. There are a range of iMGs currently available, with varying hardware and processing approaches utilized (see [9,11] for recent comparisons).

Most instrumented mouthguards reported in the literature have shown to measure head kinematics values that are in close agreement to the measurements of crash-test sensors firmly installed on anthropometric testing devices (ATD) [9,10,11,12,13]. Such comparisons are made in lab experiments that subject the ATD headform to a range of impact conditions. However, only a few studies have tested iMGs under conditions that produce peak linear accelerations (PLA) over 100 g at the center of mass of the head [10,14,15]. There is evidence that such accelerations occur in sports and that they are often related to an increased risk of injury [16,17,18,19]. For instance, dynamic changes in blood–brain barrier regulation have been reported in professional mixed martial arts fighters who sustained PLAs over 100 g and PRAs over 10 krad/s^2^ [18]. Loss of consciousness has been associated with boxing punches of 123 (±65) g [20]. Concussion has been associated with unhelmeted Australian rules impacts producing PLAs of over 100 g [21]. Finally, impacts producing PLAs of up to 146 g have been reported in concussed American Football players [17]. While these values are reported from laboratory reconstructions [20], computer reconstructions [21], or error prone helmet-based telemetry systems [17], it is clear that impacts producing high PLAs (while rare) are likely to occur. Such high magnitude impacts are most likely to cause injury and would play an important role in defining brain injury thresholds. Hence, it is crucial that iMG measurements are valid and reliable within this range of head kinematics values.

iMGs measure head kinematics, such as PLA, but head kinematics do not provide information about mechanical strain within the brain, which is shown to be linked to short- and long-term brain injuries [6,22,23,24,25,26]. Computational models of brain injury biomechanics can provide this information, but they need to be loaded with head kinematics. Hence, the reliability and validity of iMG systems should be tested for providing input data for accurate brain modeling. Previous work has found that the rotational kinematics significantly affected brain strain predicted by a range of brain models, whereas the linear acceleration did not [27,28]. It is therefore essential to test instrumented mouthguards under conditions that emphasize the rotational kinematics produced by head impacts. However, comparison testing of iMGs to date has predominately focused on linear kinematics via pendulums and pneumatic impactors [9,13,29,30]. Moreover, to the authors’ knowledge, the influence of iMG on brain strain estimations when compared to the ATD gold standard has been assessed in a single study, where a previous version of the current iMG utilizing different sensors was evaluated [11]. iMGs were assessed against a reference ATD under helmeted headform impacts simulating American Football impacts and producing PLAs from ~20 g to ~100 g and PRVs of ~15 rad/s to ~50 rad/s (absolute ranges not reported). Hence, there is a need for more studies that evaluate the performance of iMGs in predicting brain response under head kinematic magnitudes and time durations that are associated with mild TBI [26,31,32].

In this study, we evaluate an instrumented mouthguard at impact magnitudes higher than previously examined (~20 to 100 g and 1000 to 6000 rads/s^2^; ref. [11]), including those reported in head impacts leading to different types of brain injuries. We test the instrumented mouthguard in laboratory conditions that represent the linear and rotational kinematic magnitudes and durations of reported on-field events and those that produce large and complex rotational kinematics in ranges previously untested. We test whether the iMG measurements agree well with the measurements of sensors fixed to the headform in the absence of on-field biofidelic noise such as shouting and biting. In addition to using measures of head kinematics, we use a detailed finite element model of brain injury biomechanics to predict brain strain from iMG and headform sensors. We test the validity of the iMG data in predicting brain strain as a key outcome measure during sporting head impacts.

## 2. Materials and Methods

### 2.1. The Instrumented Mouthguard

The iMG presented here is version 2 of the PROTECHT system. Version 1 has been validated against ATD sensor data but under impact conditions that produced low magnitudes of head kinematics, from ~10 to 100 g and 1000 to 6000 rads/s^2^ across multiple studies [11,12,33]. Version 2 contains a tri-axial accelerometer (±400 g, 12-bit resolution, H3LIS331DL, STMicroelectronics, Genova, Switzerland) sampling at 1 kHz and a tri-axial gyroscope (±35 rad·s^−1^, 12-bit resolution, LSM6DSO, STMicroelectronics, Genova, Switzerland) (Figure 1A). The gyroscope samples at 1666 Hz, which is downsampled to 1 kHz to match linear accelerometer outputs. Sensors carry on-board filtering capabilities that are turned off for all data capture. The sensors are encased within 1.5 mm of Erkoflex and 2 mm of ethylene vinyl acetate (EVA). The mouthguard is CE and UKCA safety certified [34].

For each impact, the inertial sensors collect 104 ms of data—10 ms pre-trigger and 94 ms post-trigger. For the current data collection, the trigger-point of the sensors is a raw linear acceleration exceeding 10 g in any one of the three axes, informed by a minimum head acceleration threshold of daily activities [35]. Rotational accelerations are derived from filtered rotational velocity time-series using a five-point stencil approximation. All signals are filtered using a low-pass 4th-order Butterworth filter with a 160 Hz cutoff frequency, chosen to align with previous approaches [11], and confirmed to be appropriate for current signals through frequency content analysis. Filtered linear accelerations are then transformed to the center of mass of the head utilizing filtered rotational kinematics using a previously defined methodology [10]. A summary of the specifications and processing is outlined in Table 1. All iMG calculations were completed within MATLAB software (R2022b; Mathworks, Natick, MA, USA).

### 2.2. Testing Protocol

In order to test the mouthguard sensors, we used helmeted headform drop tests. With this approach, the impact severity and location can be easily controlled, and head kinematics across a wide range of values can be generated [37]. Testing was completed in two separate sessions, with a total of 91 impacts to the front, side, and back (25, 52, and 14 impacts, respectively). In addition, since there is no robust biofidelic dummy neck available for head impact testing, using an isolated headform avoids any spurious effects of the dummy neck on the head kinematics.

Testing was conducted via the Imperial College HEAD lab’s drop rig to deliver impact to a 50th percentile male Hybrid III headform (ATD) (Figure 1B). The headform was instrumented with an array of nine PCB piezoelectric accelerometers (PCB 352C23, PCB Piezotronics, Depew, NY, USA), mounted inside the headform in a 3-2-2-2 arrangement. This allowed for the measurement of both linear and rotational accelerations at the center of mass of the headform using Padgaonkar’s method, the same as in several previous studies [36,37,38,39,40,41]. The acceleration data were recorded at a 50 kHz frequency and filtered with a 4th order Butterworth low-pass filter with a 1 kHz cut-off frequency, with the ATD showcasing high fidelity in previous literature [42]. A dummy neck was not utilized due to the nature of the drop test set-up and the unavailability of a robust biofidelic neck suitable for headfirst impacts. The vast majority of previous studies have used the Hybrid III neck, but this neck is designed for frontal car collision tests, and it is not biofidelic under side impacts nor under direct head impacts [43]. Hence, including this neck can lead to results that are not representative of real-world human head responses.

The HIII headform does not have an upper dentition to allow for fitting the mouthguard. Hence, we fixed a plastic model of the upper dentition inside and to the top of the headform’s metal casing. The instrumented mouthguard was fitted to the plastic model, and the headform cap was fixed to the casing (Figure 1B). The headform was then fitted with a cycling helmet (the DesignSter Lightweight Helmet). The helmet provides cushioning to the headform, similar to the padding added to impactor pendulums used in previous studies, whereby these materials are utilized to modulate impact characteristics. We used the helmet to achieve a wide range of head acceleration magnitudes and durations representative of impacts in sports such as rugby and boxing [9,44,45,46], whereas under the current drop-testing conditions, an unhelmetted ATD would produce impact durations drastically shorter than those seen on the field. The helmet carries the added benefit of protecting the underlying headform from repeated impacts. The helmeted headform was lifted from the ground and dropped onto a metal anvil angled at 45° and covered with 80-grit abrasive paper (Figure 1C). Larger head rotational accelerations and longer impact durations are produced by using the oblique anvil compared with horizontal anvils.

The headform was subjected to impacts at different locations (Front, side, rear-side, and rear) and speeds (2, 3, 4, 5, 6, 7, and 8 m/s) (Figure 1D). Impact locations were chosen to represent common field head impact locations [47]. Changing the impact location changed the dominant component of linear and rotational accelerations of the headform, thus testing the iMG under a wide range of acceleration directions. In addition, changing the impact speed led to a wide range of magnitudes and durations of accelerations. We considered the accredited range of the iMG’s sensors, ±400 g and ±35 rads/s per axis for the accelerometer and gyroscope, respectively, as the limitation of sensor measurement and therefore did not include impacts that exceeded this range. For consistency, checks were performed after each impact to ensure the mouthguard had not come loose before proceeding to the next impact test.

To ensure consistency across studies, methodological and procedural information is presented in line with the recent Consensus Head Acceleration Measurement Practices (CHAMP) recommendations [48,49].

### 2.3. The Finite Element Model of the Human Brain

The Imperial College finite element model of the human head was used to predict brain strain during impacts (Figure 1E). The model incorporates details of the brain anatomy from high-resolution magnetic resonance images of a healthy 34-year-old male subject. The model consists of nearly one million hexahedral elements and a quarter of a million quadrilateral elements, representing 11 tissues, including the scalp, skull, brain, meninges, subarachnoid space, and ventricles. The details of the model development, mechanical properties of different tissues, and validation of the model predictions against sonomicrometry brain displacement data and intracranial pressure data from post-mortem human subject experiments can be found in [6,40,50,51,52,53,54]. A non-linear transient dynamic code, LS-DYNA [55], was used to set up the model and solve the equations using 20 cores of a high-performance computer and 16 GB of RAM.

The head model was loaded by the linear and rotational accelerations obtained from the ATD and iMG head impact data at the center of mass. The duration of the simulations was 30 ms, which was enough to allow strain to reach its maximum value across the brain. The maximum principal value of the Green–Lagrange strain tensor, which the element experienced during the impact, was determined (Figure 1E). The 90th percentile value of strain distribution across the whole brain was calculated, which is called the maximum principal strain (MPS) or strain hereafter.

### 2.4. Data and Statistical Analysis

For ATD and iMG comparisons, dependent variables were peak resultant linear acceleration measured at the center of mass of the headform (PLA), peak resultant rotational velocity (PRV), peak resultant rotational acceleration (PRA), and maximum principal brain strain (MPS). Headform and iMG data were time aligned such that 10 ms prior to both traces crossing a 10 g threshold was set as timepoint 0, and a total of 35 ms of the data were used to compare the time-series data.

To test the agreement between iMG and ATD measurements, we used a battery of statistical tests, namely linear regression, concordance correlation, Bland–Altman analysis, and ordinary least products regression. These methods have been used in previous studies that have validated iMG measurements [9,30]. Here, we have provided a brief explanation and logic for using these methods.

We used linear regression R-squared, which indicates the proportionate amount of variation in the response variable (e.g., the iMG data) explained by the independent variables (e.g., the ATD data) in the linear regression model [56]. R-squared values were calculated within R-Studio (R Studio Team, Boston, MA, USA). Although commonly used within iMG validation methodology [14,29], correlation and linear regression model R-squared values do not assess statistical agreement between measures if interpreted in isolation. Hence, we also used the concordance correlation coefficient (CCC), which evaluates the degree to which pairs of observations fall on the 45° line through the origin [57]. CCC values for linear and rotational kinematic measures were calculated and combined to represent the overall iMG measurement validity [30]. The minimum validity threshold value for CCC was considered 0.80, as suggested in previous studies [30,58].

We also used the Bland–Altman 95% limits of agreement analysis to establish the level of agreement between the iMG and ATD measurements. Bland-Altman analysis evaluates the mean difference (bias) between two measurement methods and estimates an agreement interval within which 95% of the differences between methods fall [59]. Visual analysis of Bland–Altman plots also offers information on whether differences are fixed or proportionate. Bland–Altman analysis was conducted using the “blandr” package [60] on RStudio (RStudio, Vienna, Austria). Differences were calculated weighing towards the ATD system, meaning positive bias indicated an underestimation in the iMG. Analyses were expressed using percentage differences and plotted using a custom RStudio script with absolute differences expressed in relation to the ATD reference. Although a priori 95% limits of agreement are usually required for Bland–Altman analysis [61], there is a lack of clinically informed criteria regarding what constitutes “agreement” within head impact sensors. Therefore, ordinary least products regression was also implemented, which assesses fixed and proportional bias between two measurement systems [62]. If the 95% confidence interval for the intercept did not include 0, then a fixed bias was present. If the 95% confidence interval for the slope did not include 1.0, then proportional bias was present. Ordinary least products regression analysis was completed in RStudio using the “lmodel2” package [63].

Mean relative errors in peaks, i.e., the average percentage difference between the ATD and iMG peak values, were also calculated. Finally, root mean-square errors (RMSE) were calculated to assess the accuracy of the overall time-series data, following a modified procedure from previous research shown in Equation (1) [10]. The RMSE errors were normalized (nRMSE) based on the impact magnitude (Equation (2)):(1)$$RMSE=\sqrt{\sum_{i}^{n}(\frac{{iMG}_{i}-{ATD}_{i}}{n}}{)}^{2}$$
(2)$$nRMSE=\frac{RMSE}{{ATD}_{max}-{ATD}_{min}}\times 100$$
where n is the number of measurements (35 ms), and ATD_max_ and ATD_min_ are the maximum and minimum values recorded by the ATD during the impact. Prior to RMSE calculations, iMG and ATD signals were trimmed and time-aligned utilizing the raw linear peak within the dominant (highest absolute magnitude) linear accelerometer axis. For RMSE calculations, trial start was defined as 10 ms prior to this raw peak (in line with the iMG 10 ms pre-sample) and trial end 25 ms after the raw peak, utilizing a longer simulation than an otherwise identical method [10].

## 3. Results

### 3.1. Impacts Produced a Large Range of Head Kinematics and Brain Strain

A total of 91 trials were collected. Ten trials were omitted from analysis and brain simulation from the front (n = 5), side (n = 1), rear (n = 3), and rear-side (n = 1), due to exceeding gyroscope measurement ranges on at least one axis of rotation. 11 of the remaining trials did not include the 10 ms of pre-impact sample time (front = 3, side = 3, rear = 5). These impacts remained within the dataset as they still allowed for the determination of peak kinematics. 81 impacts were taken forward for analysis, with durations ranging from 6–18 ms, PLA ranging from 16–171 g, PRA ranging from 1330–10,164 rads/s^2^, and PRV ranging from 11.3–41.5 rad/s, as measured by the ATD sensors. These impacts produced a large range of MPS, 0.057–0.280, as predicted by brain modeling.

Sample headform accelerations for front, side, rear, and rear-side impacts are shown in Figure 2. As can be seen, these impacts produce linear and rotational accelerations with dominant components along different axes. The front, rear, and rear-side impacts produce large linear acceleration components along the z and x axes and a large rotational acceleration component about the y axis. The side impact produces large linear acceleration components along the y and z axes and a large rotational acceleration component about the x axis. In addition, the rotational acceleration components change direction during impacts, subjecting the iMG to more complex loading conditions.

### 3.2. Good Agreement between the iMG and ATD Measurements of Peak Head Kinematics Values and Brain Strain

Figure 3 presents scatter plots and R^2^ values for PLA, PRV, PRA, and MPS for the iMG system compared to the ATD reference system. The highest R^2^ value was for PLA (0.99) and the lowest for PRV and MPS (0.94). The high R^2^ values show a strong correlation between the systems.

The values for CCC are presented in Table 2. The highest CCC was for PLA, and the lowest for PRA. The high CCC values indicate substantial agreement between the measurements of the iMG and ATD, with mean values exceeding 0.95, higher than the minimum required value of 0.8 [30].

### 3.3. Low Error in iMG Measured Peak Values and Time-Series Data Compared with ATD

The maximum difference in the peak values between iMG and ATD was for PRA, showing that on average the iMG measurements of PRA are 6.7% lower than ATD (Table 2). The lowest relative error was found for PRV at 0.8%.

We then compared the time-series data between the iMG and ATD sensors using nRMSE. Rotational acceleration showed the lowest nRMSE (1.24 ± 0.86%), and rotational velocity showed the highest (5.47 ± 2.61%) (Table 2). Minimum and maximum nRMSE ranged from 0.12 to 6.74% for PLA, 0.03 to 11.68% for PRV, and 0.01 to 6.74% for PRA. Figure 4 compares the iMG and ATD data for impacts with the most and least nRMSE for each measured kinematics variable, showing good agreement between the iMG and ATD measured time-series data.

### 3.4. Evidence of Bias in Some of the iMG Measurements

Within Bland–Altman analysis, PLA, PRV, and PRA produced bias (±95% LoA) of −3 g (−13.7–7.4 g), 0.32 rad/s (−3.6–4.3 rad/s), and 439 rad/s^2^ (−680–1566 rad/s^2^) respectively (Figure 5). Ordinary least products regression analysis found evidence of fixed (intercept = 3.707; 95% CI 1.37–5.99) and proportional (slope = 0.93; 95% CI 0.91–0.95) bias for PLA and evidence of fixed (intercept = 1.15; 95% CI 1.09–1.21) and proportional (slope = −400; 95% CI −726–−89) bias within PRA. There was no evidence of fixed (intercept = −0.65; 95% CI −2.25–0.86) or proportional (slope = 1.03; 95% CI 0.98–1.08) bias for PRV. There was no evidence of fixed (intercept = −0.0000683; 95% CI −0.0108–0.00892) or proportional (slope = 1.021; 95% CI: 0.9688–1.0762) bias for MPS.

## 4. Discussion

We presented an instrumented mouthguard that allows for measuring a broad range of head linear and rotational kinematics observed in sports. This was enabled by using tri-axial piezoelectric accelerometers and gyroscopes that have a large measurement range. We found close agreement between the instrumented mouthguard (iMG) and anthropometric test device (ATD) measurements of head linear and rotational accelerations and rotational velocity, produced by controlled helmeted head impacts at different impact locations and speeds. We also predicted brain strain as a key outcome measure from iMG and ATD data using a detailed and validated FE model of the human brain. This analysis confirmed the accuracy and reliability of iMG’s sensors for measuring head kinematics and brain strain under the simulated impact conditions.

The measured iMG kinematics data had a total correlation concordance coefficient (CCC) value of 0.970 when compared to the ATD reference measurement system, meeting minimum industry standards to progress to field testing [30]. The total CCC value reported in the current study is similar to those reported in previous literature evaluating a cohort of instrumented mouthguards [9] and similar to an older version of the current mouthguard [12]. However, testing conducted by both aforementioned studies utilized an alternative experimental set-up: the testing conducted by Jones et al. [9] used a pendulum impactor on a bareheaded ATD headform, and the testing reported by Jones and Brown [12] used a pneumatic linear impactor on a helmeted ATD headform, with both studies only achieving PLAs up to 100 g due to equipment limitations. PLAs of over 100 g have been reported in sports such as boxing, American football, and [9,17,18,19]. Such high acceleration magnitudes have an increased chance of causing traumatic brain injuries, although it should be noted that increased strain magnitudes of brain tissues are predominantly due to the increased rotational kinematics accompanying high linear kinematics [64]. As the recording and modeling of large impacts could enable researchers to better investigate causal mechanisms of injury, the validity and reliability of on-field kinematics inputs are vital, requiring iMGs to be validated within a large range of kinematics, as was conducted in this study.

Ordinary least product regression analysis showed fixed and proportional bias for both linear and rotational acceleration. Proportional bias could suggest that as the magnitude of impact increases, the amount of variability within the agreement between the ATD and iMG also increases. However, when analyzed via percentage difference, the data conformed to a homoscedastic distribution, suggesting the magnitude of error only increases in line with the magnitude of impact and not due to increasing variability. Bland–Altman analysis showed a slight overestimation of PLA within the iMG of 2.2% and an underestimation of PRA of 6.16%. Limits of agreement ranged between −14.4 and 10% (absolute 24.4% LoA) for PLA, between −13.7 and 15% (absolute 28.7% LoA) for PRV, and between −8.7 and 21% (absolute 29.7%) for PRA. There are currently no clinically meaningful criteria for what constitutes acceptable agreement with iMGs. However, these results are comparable to, and in the case of PLA, lower than, previous mouthguard validations, where the top-performing mouthguard reported 31.7% and 29.7% limits of agreement for PLA and PRA respectively [9,13]. It is acknowledged that the PRA within the current study falls below values of comparable systems, with the CCC ranging from 0.965 to 0.986 under linear impactor conditions compared to the CCC of 0.945 in the current study [9]. Such differences could potentially be due to the reduced sampling frequency of the current iMG, whereby the differentiation process introduces additional noise. However, CCCs are still well above industry standards [30].

It is important to evaluate the accuracy of the overall shape of the time-series kinematics data if such data is used for predicting brain tissue response [10]. The mean RMS (and nRMS) error for the linear acceleration and rotational velocity was 2.76 ± 1.61 g (2.8 ± 1.3%) and 1.5 ± 0.9 rad/s (5.5 ± 2.6%), respectively. The errors measured in our study are comparable with, and on a normalized basis lower than, those reported [10], where the mean RMS (and NRMS) errors for the linear acceleration and rotational velocity were 3.9 ± 2.1 g (9.9 ± 4.4%) and 1.0 ± 0.8 rad/s (10.4 ± 9.9%), respectively.

The maximum principal brain strain predicted by inputs from ATD or iMG showed high levels of agreement, with a high CCC value of 0.970 and a low mean relative error of 1.67%. Bland–Altman 95% limits of agreement analysis reports a bias of 1.35% with limits ranging between −13.85% and 16.59%, and ordinary least product regression reported no fixed or proportional bias. As previously discussed, although Bland–Altman analysis usually requires an a priori set level of agreement between the two measurements, there is currently no set level of agreement informed by clinical measures within the brain modeling literature. This study, for the first-time, provides the results of this analysis for brain strain and hopes to encourage similar approaches in the future when measuring the validity of iMGs that are used to predict brain strain via finite element models of the human brain.

The helmeted headform impacts onto the oblique anvil exposed the iMG device to PLA ranging from 16 to 171 g, PRA ranging from 1330 to 10,164 rads/s^2^, PRV ranging from 11.3 to 41.5 rad/s, and impact durations of 6–18 ms. Although the use of a helmet could be viewed as reducing generalisability to non-helmeted sports, these ranges of head kinematics represent those reported in a range of sports. Within rugby, typically PLAs and PRAs have been reported of 22.2 ± 16.2 g and 3902 ± 3948 rads/s^2^, with mean impact durations of 12 ms across all impacts [45]. In another study, true-positive impacts have been reported ranging in PLAs from 10 to 80 g and PRAs from 1000 rad/s^2^ to 8000 rad/s^2^ across varying iMGs that had attained within lab CCCs of over 0.95, but impact durations were not reported [9]. Within boxing, laboratory reconstructions of varying punches elicited a range of accelerations with short durations, from PLAs of 24.1 ± 12.5 g and 71.2 ± 32.2 g for uppercut and hook punches, respectively, to PRAs of 3181 ± 1343 rad/s^2^ and 9306 ± 4485 rad/s^2^ for uppercut and hook punches, respectively [46]. Within sports, while the majority of impacts will fall within the discussed ranges, impacts of much higher magnitude can occur and can be associated with brain injury. As previously discussed, loss of consciousness has been associated with boxing punches of 123 (±65) [20], and concussion has been associated with unhelmeted Australian rules impacts producing PLAs of over 100 g [21] and 146 g in American Football players [17]. Boxing headgear has also been tested up to PLAs of 123 ± 3 g and 8605 ± 113 rad/s^2^, indicating a need for other instrumentation intended for boxing use to be valid within these ranges [44]. Whilst these impacts are rare—with only 0.5% of observed PLAs over 106 g and 12.7% of observed PRAs over 7900 rad/s^2^ within rugby [45]—their significance with regards to injury recognition is high, and if inputs are to be fed forward to understand underlying brain motion, it is vital that sensors are valid within these ranges.

One limitation of the current study is the placement of the iMG at the top of the ATD, as the headform used could not facilitate placement within the “jaw”, similar to the procedures of previous studies [33]. While a jaw placement would emulate in vivo conditions to a greater extent, it has been noted that maintaining the position of the “jaw” within a headform is problematic due to no active mandible contraction [11]. A study has fixed a plate to the upper jaw to avoid this issue and prevent the relative motion between the iMG and upper jaw during impacts [9]. While it is acknowledged that a moving jaw introduces further sources of noise within an impact signal, the current study sought to characterize the validity of sensors and data processing procedures in the absence of these real-world noise sources. As such, the results of the current study purely indicate that the iMG shows suitable in-laboratory validity to begin testing on-field validity, where the positive predictive value of true positives is ascertained as outlined by previous research [30]. In addition, these results do not claim to validly measure in vivo impacts at this time. Within a laboratory environment, noise contributions within the signal can be contributed to signal processing procedures—such as inherent sensor noise, axis alignment, differentiation, and transformation to the head center of mass—which the current study addresses with the use of simple data filtering. On-field noise contributions—such as shouting, biting, or direct sensor impacts—may not be as easily removed and may require novel approaches to ensure the true underlying signal is retained. Future work should aim to address the identification of biofidelic noise within a controlled environment due to the difficulties of measuring a gold standard reference signal in vivo, which may require novel ATD construction or artificial signal generation. It should also be noted that some trials (n = 10) did not include a pre-sample time due to a user software error. These trials were still included within the dataset as the measurement of peak impact kinematics in all axes was still possible.

## 5. Conclusions

The current study measures the agreement between the ATD reference system and an iMG system for impacts producing head kinematics that may be representative of injurious impacts in sport, with a reported mean CCC value of 0.970. In this study, the iMG system was comparable to that of a gold standard system for a large range of PLA, PRA, and PRV and impact duration. iMG derived kinematics were also found to be comparable to gold standard kinematic inputs for finite element brain modeling, achieving a CCC of 0.97 for brain strain. While the current validation does not address true or false positive impact identification or assess biofidelic noise contributions, it does indicate that the system possesses sufficient validity for on-field use when combined with video verification of impacts. This could provide coaches or sideline medical practitioners with a useful tool for the identification of large head impacts within a live environment or be used to assess the impact demands of sports when combined with video verification of impacts. In addition, the test method presented here may provide an alternative approach for measuring the validity of iMG sensor measurements for a large range of head kinematics.

## Figures and Tables

**Figure 1 sensors-23-07068-f001:**
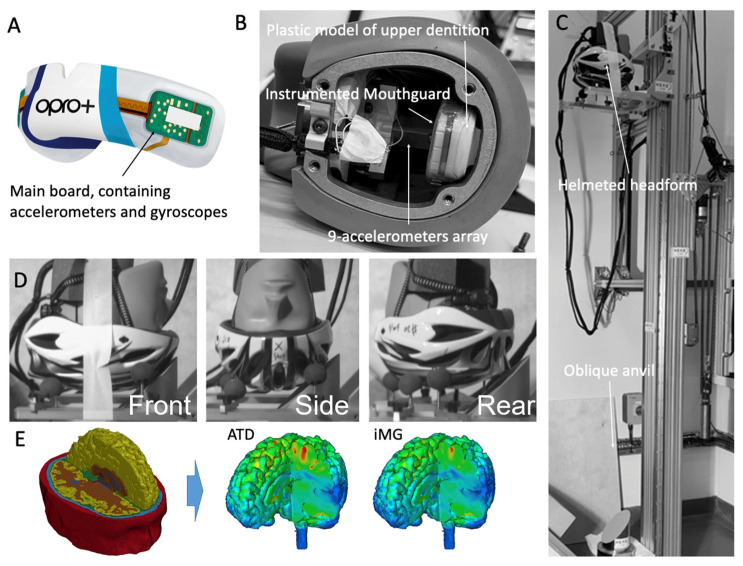
An overview of the methods. (**A**) The instrumented mouthguard; (**B**) the HIII headform, with the 9-accelerometer array installed on the metal casing and the mouthguard fitted to a plastic model of the upper dentition fixed to the metal casing; (**C**) the oblique impact helmet test setup; (**D**) different impact locations producing a range of head kinematics about the anatomical axes of the headform (left-to-right; front, side and rear); (**E**) the Imperial College Model of the Human Head and sample strain predictions using kinematics data from the headform sensors (ATD) and instrumented mouthguard sensors (iMG).

**Figure 2 sensors-23-07068-f002:**
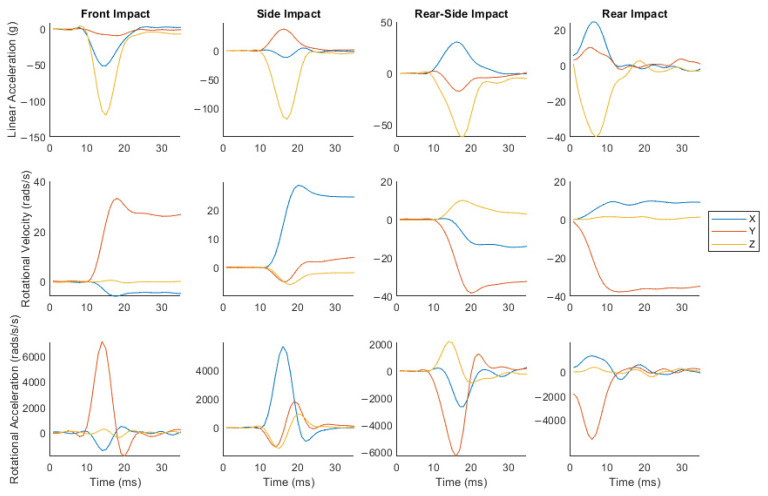
Sample headform accelerations from front, side, rear-side, and rear impacts. These graphs show that these impacts produce a range of dominant components of the linear and rotational accelerations. Note that for rear impact trials, there was no pre-impact sample available for linear or rotational kinematics.

**Figure 3 sensors-23-07068-f003:**
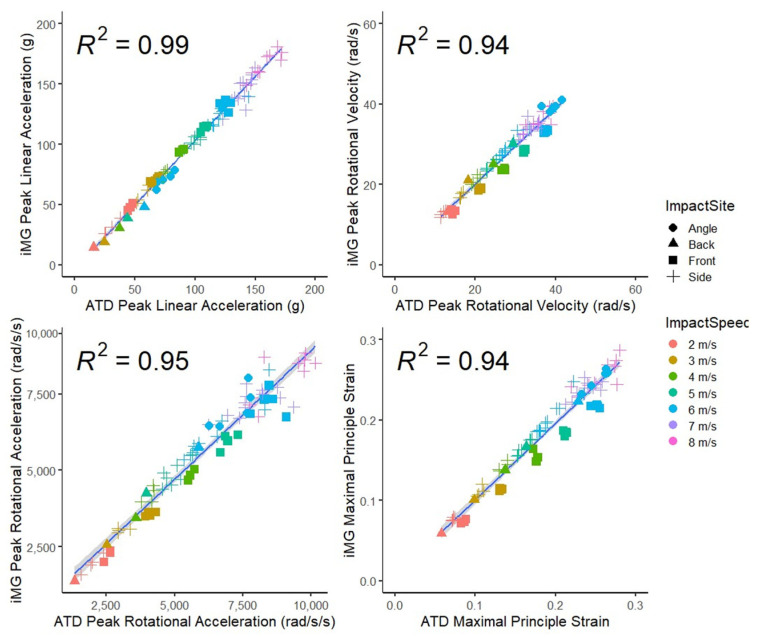
Scatter plots for peak linear acceleration, peak rotational velocity, peak rotational acceleration, and maximal principal strain, measured by the ATD and iMG. Linear regression trendline (±95% CI on line) and R squared values are displayed on each graph.

**Figure 4 sensors-23-07068-f004:**
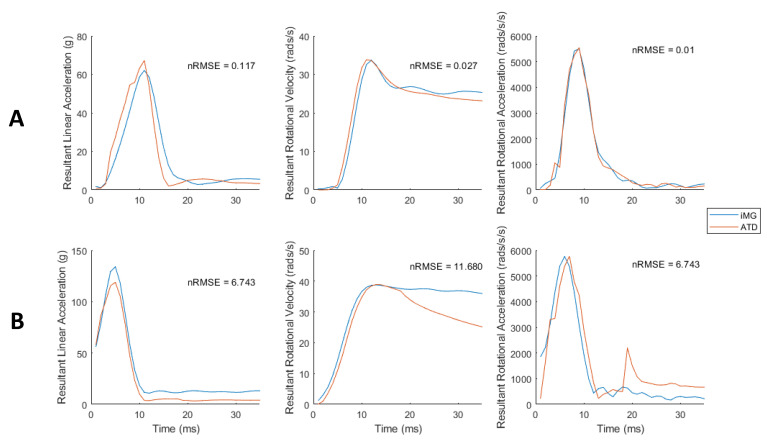
Resultant kinematics of iMG and ATD vs. time for impacts that produced the lowest (**row A**) and highest (**row B**) nRMSE (%) for each kinematics measure.

**Figure 5 sensors-23-07068-f005:**
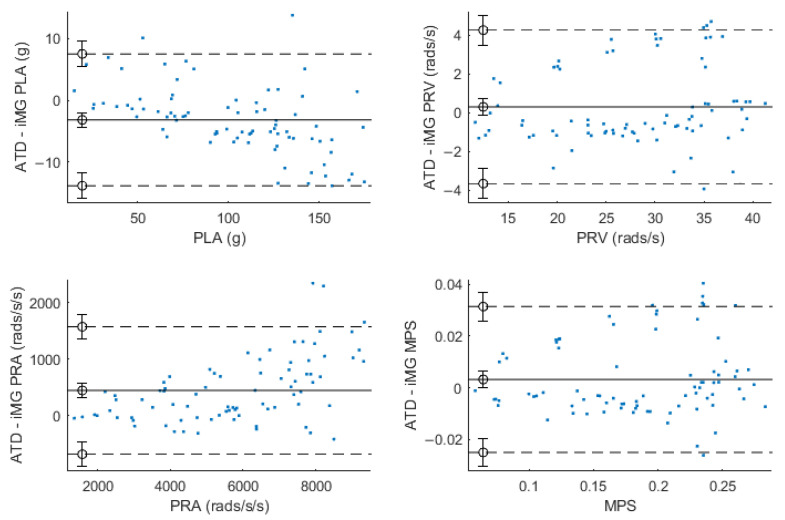
Bland–Altman plots for PLA (**top left**), PRV (**top right**), PRA (**bottom left**), and MPS (**bottom right**). Solid lines refer to bias, dashed lines refer to 95% limits of agreement.

**Table 1 sensors-23-07068-t001:** Specification and processing of the instrumented mouthguard and ATD systems.

	iMG (PROTECHT^TM^ System)	ATD (Reference)
Sampling rate (Accelerometer)	1000 Hz	50,000 Hz
Sampling rate (Gyroscope)	1666 Hz and downsampled to 1000 Hz	NA
Measurement range and resolution (Accelerometer)	±400 g per axis	±1000 g
Measurement range and resolution (Gyroscope)	±35 rad/s per axis	N/A
Output time windows	[−10, 94] ms	[−2, 48] ms (after data processing)
Output coordinate axes direction	Not standard but transformed to ISO reference frame.	ISO reference frame
Output coordinate origin	Sensor which is then transformed to Centre of Mass	Centre of Mass
Filter	Post analysis identified a 160 Hz low pass 4th order Butterworth filter	4th order Butterworth low-pass filter with a 1 kHz cut-off frequency
Derivation of rotational acceleration	5-point stencil derivative	Padgaonkar’s method [36]

**Table 2 sensors-23-07068-t002:** Agreement statistics for biomechanical metrics: Concordance correlation coefficients (CCC), RMSE, nRMSE, and Bland Altman Statistics for PLA, PRV, and PRA in the ATD and iMG.

	CCC	Mean Relative Error (±SD) %	RMSE	nRMSE	Bland-Altman (% Difference)		
	(95% CI)		(±SD)	(±SD) %	Bias (95% CI)	Lower Limit	Upper Limit
PLA (g)	0.989	5.2	2.76	2.88	−2.16	−14.43	10.10
	0.984–0.992	(±4.4)	(±1.61)	(±1.27)	(−4.51–0.19)		
PRV (rad/s)	0.970	0.8	1.52	5.47	0.69	−13.74	15.12
	0.953–0.980	(±7.4)	(±0.87)	(±2.61)	(−2.07–3.46)		
PRA (rad/s^2^)	0.945	6.7	78.27	1.24	6.16	−8.72	21.04
	0.921–0.962	(±8.2)	(±71.97)	(±0.86)	(3.31–9.02)		
MPS	0.970	1.67	N/A	N/A	1.35	−13.85	16.59
	0.955–0.981	(±8.1)			(−1.59–4.29)		

## Data Availability

Data are available upon request from the corresponding authors.

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
