# Peer review of "An Instrumented Mouthguard for Real-Time Measurement of Head Kinematics under a Large Range of Sport Specific Accelerations"

_sensors, 2023, doi:10.3390/s23167068_

Round 1
Reviewer 1 Report
This is an interesting study that compared peak kinematic measurements between an ATD headform and and instrumented mouthguard under various oblique drop testing conditions. The authors expanded on this work by assessing mouthguard sensor performance using multiple statistical metrics and also examining the corresponding levels of brain strain through a validated finite element model. This work is significant given the adoption of mouthguard-based sensors to further concussion research in many non-helmeted sports. However, there are some minor revisions that need to be made before this manuscript is ready for publication. Recommendations are provided below.
Page 2, lines 93-94
Although the Lieu et al. 2020 study is already cited, it would be beneficial to specify the range of impact magnitudes in this sentence to facilitate future study comparisons.
Page 3, lines 106-107
I recommend defining the low magnitudes of head kinematics here even though multiple studies are referenced.
Page 3, line 14
Insert a few references that describe CE and UKCA safety certifications so that readers outside Europe can research these standards if they are unfamiliar with them.
Page 3, line 117
Can you briefly mention the basis for the 10 g trigger in any one of the three axes? Is it based on an internal dataset for this particular instrumented mouthguard or based on other studies that make this recommendation?
Page 3, lines 124-125
It seems there was an error and the table number was not updated properly to Table 1.
Page 4, line 133
I recommend adding a sentence towards the beginning in which you list out the number of impact locations, speeds, trials, etc. and then provide a total number of tests conducted. Also, were any locations struck more than once?
Page 4, lines 140-141
I believe you are fine to remove the phrase in parentheses (also called an anthropometric testing device, ATD, in this paper) because the term ATD is already defined in both the Abstract and Introduction.
Page 3, figure 1
I suggest adding text boxes that read "Front", "Side", and "Rear-Side" to the three test setup photos in Section D. The suggested names for these different oblique drop test impact locations match the location names used in Figure 2.
Page 5, lines 165-166
Can you clarify the statement "The helmet provides cushioning to the headform, similar to the padding added to impactor pendulums used in previous studies"? Like most bicycle helmets, the DesignSter is composed of expanded polystyrene while the impactor faces used in other studies (Kieffer et al. 2020, Jones et al 2022) are composed of nylon or vinyl nitrile padding. Although these materials differ, are you just trying to say that these materials can be used to modulate the magnitude and duration of the headform acceleration without headform surface being directly impacted during testing?
Page 5, line 174
Include the impact locations in parentheses similarly to how you defined the speeds in parentheses.
Page 8, lines 264-266
Were all 11 trials without the 10 ms pre-trigger from the Rear impact tests only, or a mixture? And for the 10 trials removed, were they from a single impact location or a mixture?
Page 10, lines 302 and 306
Table numbers need to be inserted.
Page 13, lines 418-432
In this section, there needs to be a brief discussion on the Rear impact tests noted in Figure 2 and why the pre-trigger data was unavailable compared to those from the other three impact locations. Does the instrumentation arrangement in the mouthguard cause certain locations to be more problematic when it comes to data collection? Or does mounting the mouthguard inside the Hybrid III headform near the crown and posterior skullcap lead to pre-trigger issues when you hit the Rear location?
Reviewer 2 Report
General comments:
This paper presents an instrumented mouthguard (iMG) that can measure head kinematics under a large range of linear and rotational accelerations, and compares its performance with an anthropometric testing device (ATD) and a finite element model of the human brain. The paper is well-written, clear and concise and provides relevant background information and references. The methods are described in sufficient detail, and the results are presented in an appropriate manner. The paper makes a valuable contribution to the field of head impact biomechanics and brain injury prevention, as it validates an iMG system that can be used in real-world settings to monitor head impacts and predict brain strain.
Specific comments:
1. The title of the paper could be more informative and specific, as it does not mention the application of the iMG system to sports or the comparison with ATD and brain model. A possible alternative title could be: “Validation of an instrumented mouthguard for measuring head kinematics and predicting brain strain in sports impacts”.
2. The abstract could be improved by adding a sentence about the motivation and significance of the study, such as: “This study aims to validate an iMG system that can be used in real-world settings to monitor head impacts and predict brain strain, which are key factors for understanding and preventing brain injuries in sports.”
3. The testing protocol could explain why the helmet was used and how it affected the head kinematics and brain strain. It could also justify the choice of impact locations, speeds and angles, and compare them with previous studies or real-world scenarios. It could also mention the limitations of using an isolated headform without a neck or a body.
4.The data and statistical analysis section could provide more information about how the ATD and iMG data were aligned. It could also explain why 10 ms of pre-trigger data were used for both systems and how this affected the comparison. It could also describe how the brain model was validated and calibrated, and what assumptions or uncertainties were involved.
5. Please check Line 124 “ Error! Reference source not found ”
6. The discussion section could provide more insights into the implications and applications of the study, such as how the iMG system can be used in sports to monitor head impacts and predict brain strain, what benefits or challenges it may bring to athletes, coaches, clinicians, researchers, etc., what future work or improvements are needed, etc.
7. The discussion section could discuss the potential sources of error or bias in the iMG measurements, such as sensor noise, calibration, alignment, filtering, transformation, etc., and how they could be minimized or corrected in future work.
